# Polyphenols Attenuate Highly-Glycosylated Haemoglobin-Induced Damage in Human Peritoneal Mesothelial Cells

**DOI:** 10.3390/antiox9070572

**Published:** 2020-07-01

**Authors:** Carolina Sánchez-Rodríguez, Concepción Peiró, Leocadio Rodríguez-Mañas, Julián Nevado

**Affiliations:** 1Faculty of Biomedical and Health Sciences, Universidad Europea de Madrid, Villaviciosa de Odón, 28670 Madrid, Spain; 2Department of Pharmacology, School of Medicine, Instituto de Investigaciones Sanitarias IdiPAZ, Universidad Autónoma de Madrid, 28029 Madrid, Spain; concha.peiro@uam.es; 3CIBER of Frailty and Healthy Aging (CIBERFES), Department of Geriatrics, Hospital Universitario de Getafe, 28905 Madrid, Spain; leocadio.rodriguez@salud.madrid.org; 4Genomic and Molecular Nephropathy Sections, Instituto de Genética Médica y Molecular (INGEMM), IdiPaz-Hospital Universitario La Paz, y Centro de Investigación Básica en Red de Enfermedades Raras (CIBERER), 28046 Madrid, Spain; jnevado@salud.madrid.org

**Keywords:** polyphenols, Amadori adducts, apoptosis, highly glycosylated haemoglobin, human mesothelial cells, reactive oxygen species, pro-inflammatory gene

## Abstract

We investigated the cytoprotective role of the dietary polyphenols on putative damage induced by Amadori adducts in Human Peritoneal Mesothelial Cells (HPMCs). Increased accumulation of early products of non-enzymatic protein glycation—Amadori adducts—in the peritoneal dialysis fluid due to their high glucose, induces severe damage in mesothelial cells during peritoneal dialysis. Dietary polyphenols reportedly have numerous health benefits in various diseases and have been used as an efficient antioxidant in the context of several oxidative stress-related pathologies. HPMCs isolated from different patients were exposed to Amadori adducts (highly glycated haemoglobin, at physiological concentrations), and subsequently treated with several polyphenols, mostly presented in our Mediterranean diet. We studied several Amadori-induced effects in pro-apoptotic and oxidative stress markers, as well as the expression of several pro-inflammatory genes (nuclear factor-kappaB, NF-kB; inducible Nitric Oxide synthetase, iNOS), different caspase-activities, level of P53 protein or production of different reactive oxygen species in the presence of different polyphenols. In fact, cytoprotective agents such as dietary polyphenols may represent an alternate approach to protect mesothelial cells from the cytotoxicity of Amadori adducts. The interference with the Amadori adducts-triggered mechanisms could represent a therapeutic tool to reduce complications associated with peritoneal dialysis in the peritoneum, helping to maintain peritoneal membrane function longer in patients undergoing peritoneal dialysis.

## 1. Introduction

Peritoneal dialysis (PD) is an effective alternative method to haemodialysis in patients with nephropathy and end-stage renal disease (ESRD) [1]. During PD, peritoneal dialysis fluids (PDF) have shown cytotoxic effects based on their high glucose and lactate concentrations, the acidic pH, and numerous toxic glucose degradation products (GDP) [2,3]. High concentrations of glucose impair numerous cellular functions [4,5], including inducing non-enzymatic glycation of cell surface and interstitial matrix proteins. Non-enzymatic glycation involves the condensation reaction of a sugar with a protein, resulting first in the rapid formation of a Schiff base, after which these adducts then undergo rearrangements to stabilize the Amadori adducts. Only a small portion of these Amadori-adducts undergoes further irreversible chemical reactions leading to the formation of advanced glycation end-products (AGEs) [6,7]. The Amadori adducts or AGE transformation of proteins on the peritoneal barrier, which is formed by mesothelial cells (MCs), induces loss of ultrafiltration capacity and increases membrane permeability to low-molecular-weight solutions [8]. 

Over the last decade, we have shown that Amadori adducts, such as high glycated human haemoglobin (HHb) and glycated bovine serum albumin (gBSA) may favour a pro-inflammatory and pro-apoptotic state (called damage) in several cell types, such as aortic smooth, endothelial and MCs [9,10,11,12]. Such damage effects, which involve reactive oxygen species, occur secondary to exacerbated activity of the nitric oxide synthase (NOS) pathway with enhanced expression of several nuclear factor-kB (NF-kB)-related proinflammatory genes [9,10,11,12]. Indeed, our data strongly support the idea that long-term exposure of the peritoneum to high concentrations of glucose solutions during PD maintains the peritoneum in a state of hyperglycaemia. This could act as an inducer of apoptosis in MCs through Amadori adducts, involving different oxidative (ROS) and nitrosative (RNS) reactive species [13,14]. Interference in this stress-induced environment may help to understand how to extend PD in our patients.

Among different possible experimental approaches, the study of the effect of several polyphenols has been reached in the last decade, due to i) a wide range of biological actions including their anti-oxidant, anti-inflammatory, anti-cancer, and anti-microbial properties; and ii) their great abundance in our Mediterranean diet and their role in the prevention of various diseases. In fact, continuous and optimal intake of polyphenols has been associated with lower incidence of cancer, inflammation, obesity, diabetes, cardiovascular disease (CVD), cognitive disorders and aging [15,16,17,18,19,20]. 

The mechanisms of action of polyphenols may be direct or indirect and have been reported on their:-anti-oxidant actions (such as scavenging radicals and metal ion chelation ability) [21].-anti-inflammatory actions (through inhibition of the redox-sensitive transcription factors, such as NF-kB and/or inhibition of “pro-oxidant” enzymes like inducible Nitric Oxide synthetase (iNOS), COX, XO); induction of antioxidant enzymes, as glutathione S-transferase (GST and SOD) [22,23] or acting as scavengers of reactive oxygen (ROS) and nitrogen (RNS) species and as chelators of iron and copper, causal factors strictly correlated to inflammatory diseases [24,25,26,27,28,29,30].-anti-glycation actions, mainly due to their antioxidant actions and carbonyl trapping functions [31,32,33,34,35,36].

In the present work, we study the effect of treatment with different polyphenols on oxidative stress, apoptosis and inflammation induced by Amadori adducts in Human Peritoneal Mesothelial Cells (HPMCs). Altogether, our results indicate that apoptosis of HPMCs induced by Amadori adducts is inhibited with exposure to polyphenols, and may be associated with mobilization of anti-oxidative and anti-inflammatory repair mechanisms.

## 2. Materials and Methods 

### 2.1. Cell Culture 

Human Peritoneal Mesothelial Cells, HPMCs, were isolated from omental tissue from 10 persons (Appendix A) undergoing non-urgent, non-septic abdominal surgery, using a previously described method [9]. HPMCs were cultured in Dulbecco’s modified Eagle’s medium (DMEM) containing 10% foetal bovine serum (FBS; Gibco) and 1% penicillin/streptomycin (Sigma-Aldrich). Cell cultures between passages two and five were used. The morphologic and immunofluorescence-staining features of the cells remain stable during these passages (Appendix A). 

### 2.2. Preparation of Amadori Adducts 

Commercial lyophilized non-enzymatically glycated human haemoglobin, containing either normal (5.4%) HbA 1 level (NHb) or high (11.1%) HbA 1 level (HHb), was purchased from Sigma Aldrich. Glycated solutions were prepared as previously described [9] (see Appendix A]). The concentration of glycated solutions used, HHb and NHb 10 nM, is in the physiological range [37]. 

### 2.3. Polyphenols Administration

In this study, we evaluated the effects of natural, dietary, and common polyphenols on HPMCs. We select these dietary polyphenols by focusing on their antioxidant, anti-inflammatory, and anti-glycation activities. HPMCs were treated with four polyphenols: tannic acid (10 μM), resveratrol (12.5 μM), quercetin (10 μM), and gallic acid (10 μM) (Sigma Aldrich). The selection of polyphenol concentrations was based on previous studies [38,39,40,41,42,43]. In fact, the dose of polyphenols used was shown to be non-toxic for cells, while other concentrations, of 15, 20, and 50 μM, induced cytotoxicity in HPMCs (data not shown). Polyphenols solutions were prepared according to the manufacturer’s instructions. 

Regarding the four polyphenols used, resveratrol is a polyphenolic phytoalexin belonging to the stilbene family. It is a natural dietary plant compound that occurs mainly in grape skin and seeds but is also found in wines and various other types of plant foods, especially peanuts, berries, and tea [44]. Resveratrol reveals a wide range of biological properties including anti-glycation, antioxidant, anti-inflammation, neuroprotective, anti-cancer, and anti-aging activity in various in vitro and in vivo experimental models [45,46,47,48,49]. Quercetin is a great representative of polyphenols, flavonoids subgroup, flavonols. Its main natural sources in foods are vegetables such as onions, the most studied quercetin-containing foods, and broccoli; fruits (apples, berry crops, and grapes); some herbs; tea; and wine. It is known for its antioxidant activity in radical scavenging [50]. In addition, quercetin can efficiently inhibit glycation of DNA [51] as well as suppress α-dicarbonyl compound-induced protein glycation [52,53], and possesses an anti-inflammatory potential that can be expressed on different cell types, including human ones [54,55,56,57,58,59]. Gallic acid is found in some plants such as grape seed, berries, rosaceous fruits, onion and tea leaves (green and black tea). Gallic acid possesses strong free radical scavenging, antioxidant [60], anti-inflammatory, antibacterial, antiviral [61,62,63], anticancer, anti-apoptotic [64,65] and anti-glycation [66] activities. Finally, tannic acid is a specific form of tannin, which is widely distributed in green tea, wine, grapes, and plants. It has strong antioxidant, anti-inflammatory, anticarcinogenic and antimutagenic activities [67] and anti-glycation activities [68,69].

### 2.4. Intracellular Reactive Oxygen Species (ROS) Detection

To determine the presence of intracellular ROS, HPMCs seeded onto 96-well microplates were grown to confluence and then serum-deprived for 24 h (0.5% FBS). HPMCs were then subjected to the different treatments, and 2 h later the fluorescent probe dihydroethidine (DHE, Molecular Probes) 30 μM, was added to cell cultures for 90 min at 37 °C. Once inside the cell, DHE is able to fluoresce in the presence of intracellular ROS. Intracellular fluorescence was then quantified using a microplate reader GENios Plus (TECAN), using a previously described method [10]. In these and other experiments, the Tempol (100 μM) is used like a cell-permeable nitroxide that acts as a free radical scavenger and nitric oxide spin trap.

### 2.5. Determination of Caspases-3/7 

For the quantitation of caspase-3/7 activity, a luminescent Caspase-Glo-3/7 Assay (Promega) was used according to the manufacturer’s instructions and previously described method [13]. Luminescence is proportional to the amount of caspase activity present and, for the purpose of comparison, 10 µg total protein was utilized. Luminescence was measured using a microplate reader GENios Plus (TECAN). Z-DEVD-FMK (100 μM), a specific caspase-3 inhibitor, was also used.

### 2.6. Quantitation of ATP-Levels

The CellTiter-Glo^®^ Luminescent Cell Viability Assay is a homogeneous method based on quantitation of the Adenosine triphosphate (ATP) present, which signals the presence of metabolically active cells. Luminescence is proportional to the amount of ATP present and, for the purpose of comparison, 10 µg total protein was used to normalize results. ATP was measured following the manufacturer’s protocol and previously described methods [70,71]. Luminescence was measured using a microplate reader GENios Plus (TECAN). 

### 2.7. Annexin V Detection by Indirect Immunofluorescence

Immunofluorescence examination was performed as described previously [13,72]. Briefly, HPMCs cultures were fixed with 4% paraformaldehyde (PFA) and blocked for 1 h at 37 °C. Then, samples were incubated overnight at 4 °C with a rabbit polyclonal anti-annexin V antibody (1/75; Abcam). The following cells were incubated with secondary Alexa Fluor 546-conjugated goat anti-rabbit antibody (1/250; Molecular Probes) for 45 min at 37 °C. Digital fluorescence microscopy was performed and evaluated under an Olympus BX51 fluorescence microscope (Olympus). Fluorescence intensity was quantified using the image analysis of software ImageJ (NIH). The specificity of the immunostaining was evaluated by the omission of the primary antibody (negative controls).

### 2.8. Reporter Plasmids

The reporter plasmids p5xNF-kB-Luc and luciferase-based reporter plasmid corresponding to the 50-flanking regulatory regions of human iNOS (7.2 hiNOS-luc) were purchased from Stratagene, USA.

### 2.9. Transient Transfection

Transient transfection experiments were performed as we have previously described in Peiró et al. 2003 [11]. Briefly, HPMCs (1 × 10^5^ cells) were grown to 80–90% confluence. The transfection mixture was added to cell cultures for an additional 18–20 h. The transfection mixture consisted of 2 μg of the above-mentioned plasmids incubated with 75 μL of DMEM and 7.5 μL of Superfects (Quiagen Gmbh, Germany) in vehicle medium, following the manufacturer’s instructions. Following treatment with the specified agents, HPMCs were harvested and lysed with passive lysis buffer (1×; Promega, USA), followed by one freeze/thaw cycle. The extracts were centrifuged and assayed with a luciferase reporter system (Promega, USA). Luciferase activity was expressed as relative luciferase units (RLUs) [11]. Results are normalized to protein content (1 μg).

### 2.10. Determination of p53 Levels 

The p53 protein levels were measured in cell lysates using p53 commercial enzyme-linked immune sorbent assay (ELISA) kits (Diaclone). After the development of the colorimetric reaction, the absorbance was measured at 450 nm using a microplate reader (GENios Plus, TECAN). The levels of p53 were normalized to protein content (5 μg).

### 2.11. Statistical Analysis

Results were expressed as mean ± SD (standard deviation). Statistical significance (*p* < 0.05) was evaluated by factorial analysis of variance (ANOVA) with post hoc test, using the StatView statistics program (Abacus Concepts Inc., Berkeley, CA, USA).

## 3. Results

### 3.1. Polyphenols Decrease Amadori-Induced ROS 

The ability of polyphenols to inhibit intracellular ROS was assayed by measuring fluorescent probe DHE in HPMC. Figure 1 shows that HHb (10 nM, physiological concentrations) significantly stimulates DHE fluorescence compared to baseline (B) or the presence of normally glycated haemoglobin form (NHb, 10 nM). Interestingly, increased HHb-induced fluorescence was clearly reduced (by 50–60%) in the presence of the different polyphenols used, in an analogous way to that of the intracellular superoxide anionic scavenger Tempol (Figure 1). No significant differences in antioxidant activity were observed between the different polyphenols. 

### 3.2. Polyphenols Decrease iNOS Gene Expression and NF-kB-Dependent Transcription Induced by Amadori Adducts in HPMCs

Next, we tested the ability of different polyphenols to reduce Amadori adducts-induced pro-inflammatory gene expression. As we previously reported, the 24 h HHb-stimulated iNOS promoter activity in HPMCs [9] was prevented in an appreciable way by different polyphenols (Figure 2A). Resveratrol and gallic acid may seem to have a more significant inhibitory effect on the iNOS promoter that quercetin and tannic acid. Tempol (100 μM) was used as a control inhibiting the action of HHb on the iNOS promoter. Similarly, Figure 2B shows that these polyphenols similarly inhibited HHb-induced NF-kB transcription-dependent in the HPMCs.

### 3.3. Polyphenols Block Amadori-Induced Apoptosis in HPMCs

Figure 3A shows that polyphenols treatment is sufficient to cause a significant decrease in caspase-3/7 activation after 24 h HHb exposure. No significant differences in the antiapoptotic action of different polyphenols used were denoted. As a control, HHb-induced caspase-3/7 activation was almost blunted by a specific caspase-3 inhibitor, ZDEVD-FMK (100 μM), and the ROS scavenger, Tempol (100 μM), supporting the specificity of such effects (Figure 3A). We also measured ATP levels (as an indicator of metabolic cell status) in whole protein HPMCs extracts. As expected, levels of ATP in HPMCs treated with polyphenols were significantly higher than in HPMCs treated with HHb (Figure 3B).

### 3.4. Polyphenols Decrease Levels of Amadori Adducts-Induced Annexin V in HPMCs

Treatment of the HPMC cultures with HHb shows an increase in fluorescence, which is associated with an increase of annexin-V, in the cells with respect to the polyphenol treatment groups (Figure 4) at 24 h. After treatment with polyphenols, fluorescence intensity decreases in HPMCs with HHb, therefore indicating a reduce apoptosis (Figure 4).

### 3.5. Polyphenols Decrease Tumour Suppressor Protein p53 Levels Induced by Amadori Adducts 

Tumor suppressor p53 has been shown to modulate apoptosis. We examined the effect of HHb on the protein levels of p53 in HPMC. As expected, HPMCs treated with HHb showed significantly increased p53 levels after 24 h of incubation (1.25 ± 0.1, p ≤ 0.05) (Figure 5). The results demonstrated that, compared with the HHb group, all polyphenols tested significantly decreased HHb-induced p53 levels (Resv, 1.02 ± 0.1; Quer, 1.09 ± 0.24; Tan, 1.09 ± 0.14; and Gal, 1.04 ± 0.1) (Figure 5). A concentration of 25 mM glucose (HGl, High Glucose) was used as a p53 stimulation control (1.45 ± 0.06 times induction on the basal). 

## 4. Discussion

Previous studies by our laboratory and others indicate that the Amadori adducts, such as HHb, could induce effects in vascular and mesothelial cells similar to those induced by AGEs, such as to bring about endothelial dysfunction [15], and activate different proinflammatory and/or pro-apoptotic markers in HPMCs [9,13]. We proposed that the use of different diet polyphenols might protect mesothelial cells against Amadori adducts cytotoxicity, and exert significant protection against apoptosis, oxidative stress, and inflammation. These polyphenols, based on all effects presented, must have a beneficial role for patients undergoing PD, although additional in vivo experiments may have to verify this hypothesis. However, several facts may support this idea, Amadori-albumin has been demonstrated in the glomeruli of patients with diabetic nephropathy, and the degree of staining was increased with the severity of tissue damage [73]. These findings are consistent with a role of Amadori-albumin in the development of nephropathy. It is worth noting that early glycated proteins can be incorporated into mesothelial cells by a transcytosis mechanism [74]. In addition, our previous data pointed to long-term hyperglycaemia acting as an inducer of apoptosis in HPMCs through Amadori adducts, involving different oxidative and nitrosative reactive species [9,11,12,13] 

Polyphenols constitute one of the most numerous secondary metabolites present in all commonly consumed dietary vegetables [75,76,77,78]. Polyphenols are reported to have numerous health benefits [75,76,77,78]. We have herein delineated the mechanisms by which polyphenols inhibit apoptosis in HPMCs, involving a significant decrease of oxidative stress induced by Amadori adducts. We also hypothesize that one of the protective mechanisms of polyphenols is that they could be oxidized by a direct reaction with free radicals. During this reaction, the free radicals are stabilized, and the active oxygen species are directly eliminated. In fact, polyphenols’ ability to inhibit the protein glycation are correlated to their free-radical scavenging capacity, suggesting that the inhibitory mechanism is, at least partly, due to their antioxidant property [79]. In addition, there exists the possibility that these plant polyphenols directly bind to proteins, as has been suggested [80], inhibiting the binding of sugars to these protein substrates and suppressing the promotion of the Maillard reaction. For instance, the interactions between tannins and proteins are associated with hydrogen bonding, hydrophobic interactions based on van der Waals forces, and the repelling of water, and polymerization generally intensifies the tannic effects [81]. The involvement of the carboxyl group and its steric structure, as well as the degree of polymerization, likely inhibits the Maillard reaction [81]. 

Additionally, our data also point out that polyphenols exert anti-inflammatory effects through molecular mechanisms such as inhibition of the redox-sensitive transcription factor NF-kB and inhibition of “pro-oxidant” enzyme iNOS, at least in molecular gene expression. It is widely known that polyphenols exert the anti-inflammatory action by different mechanisms: radical scavenging, metal chelating, NOS inhibition, inhibition of certain enzymes involved in ROS production, and upregulation of endogenous antioxidant enzymes [42,75,76,77,80]. This work is in agreement with previous studies, which support the preventive effects of polyphenols on inflammation in different diseases [39,48,82,83,84], including the kidney in animal models [85] or in vivo models of study, where active compounds extracted from extra olive oil (oil phenols) counteract mesothelial to mesenchymal transition of HPMCs exposed to conventional peritoneal dialysate in vitro and in vivo, preventing and counteracting the development of peritoneal fibrosis during PD [86,87]. These data are consistent with the results obtained by other groups, for example, when immortalized human mesothelial cells (Met5A) were exposed either to the regular growth medium, to standard acidic lactate-buffered PDF (Dianeal PD4), or to a more biocompatible lactate-bicarbonate-buffered PDF (Physioneal 40). In the presence of quercetin, which induced cytoprotection, potentially based on its antioxidant effects in this model [88]. Chong-Ting et al. (2016) demonstrated that supplementation with trans-resveratrol improved ultrafiltration in PD patients, and high-dose supplementation may improve ultrafiltration by ameliorating angiogenesis induced by conventional lactate-buffered PD solutions [89]. In another study, Firuzi et al. (2016) concluded that silymarin could be considered a complementary therapy to reduce PD-related complications in patients with ESRD undergoing PD. Silymarin is a standardized extract of the herb milk thistle, *Silybum marianum*. It consists of flavonolignans (polyphenols) including silybin, silydianin, and silychristine. Of these, silybin is the component with the highest biological activity [90]. Other groups observed that the decrease in inflammatory cytokines delayed the onset of diabetic nephropathy [91,92].

On the other hand, results improving the biocompatibility of peritoneal dialysis fluids (PDF) blocking oxidation and glycation on those PDFs, clearly reduce cell damage of glucose degradation products in PD patients [93]. The results presented herein in our study, an in vitro model, are strengthened by different studies that demonstrated the beneficial effects of polyphenols in patients receiving PD.

## 5. Conclusions

In conclusion, our study showed that cytoprotective agents such as polyphenols may represent an alternate approach to protect mesothelial cells from the cytotoxicity of Amadori adducts. Future studies will be needed to elucidate the in vivo effects of this approach. Therefore, interfering with the signaling mechanisms triggered by Amadori adducts could represent a therapeutic tool to reduce complications associated with exposure to PDF in the peritoneum. Indeed, polyphenols may help to maintain peritoneal membrane function longer in patients undergoing PD. Regular, daily consumption of polyphenols, in the form of food or extracts, may aid in the prevention or inhibition of non-enzymatic amino acid glycation in the living body. Non-enzymatic amino acid glycation may be associated with ageing, diabetic complications, and other diseases.

## Figures and Tables

**Figure 1 antioxidants-09-00572-f001:**
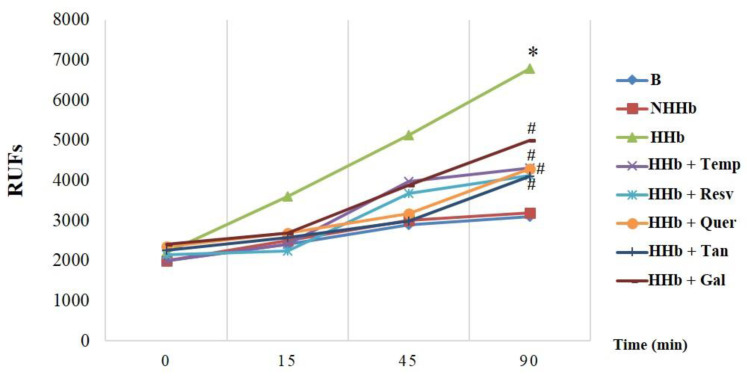
Polyphenols decrease reactive oxygen species induced by Amadori adducts. Representative experiments of the intracellular detection of superoxide anions over time using the fluorescence probe dihydroethidine (DHE). Human Peritoneal Mesothelial Cells (HPMCs) were treated with: HHb (10 nM); NHb (10 nM); HHb (10 nM) + resveratrol (Resv, 12.5 μM); HHb (10 nM) + tannic acid (Tan, 10 μM); HHb (10 nM) + quercetin (Quer, 10 μM); HHb (10 nM) + gallic acid (Gal, 10 μM); or HHb (10 nM) + Tempol (Temp, 100 μM), a Reactive Oxygen Species (ROS) scavenger. For each experiment, 10 independent cultures of HPMCs were used, corresponding to 10 donors. Data represent the mean ± SD of at least seven independent experiments (for each culture of HPMCs corresponding to each of the 10 donors) expressed as relative fluorescence units (RFUs). * *p* ≤ 0.05 vs. basal. # *p* ≤ 0.05 vs. HHb.

**Figure 2 antioxidants-09-00572-f002:**
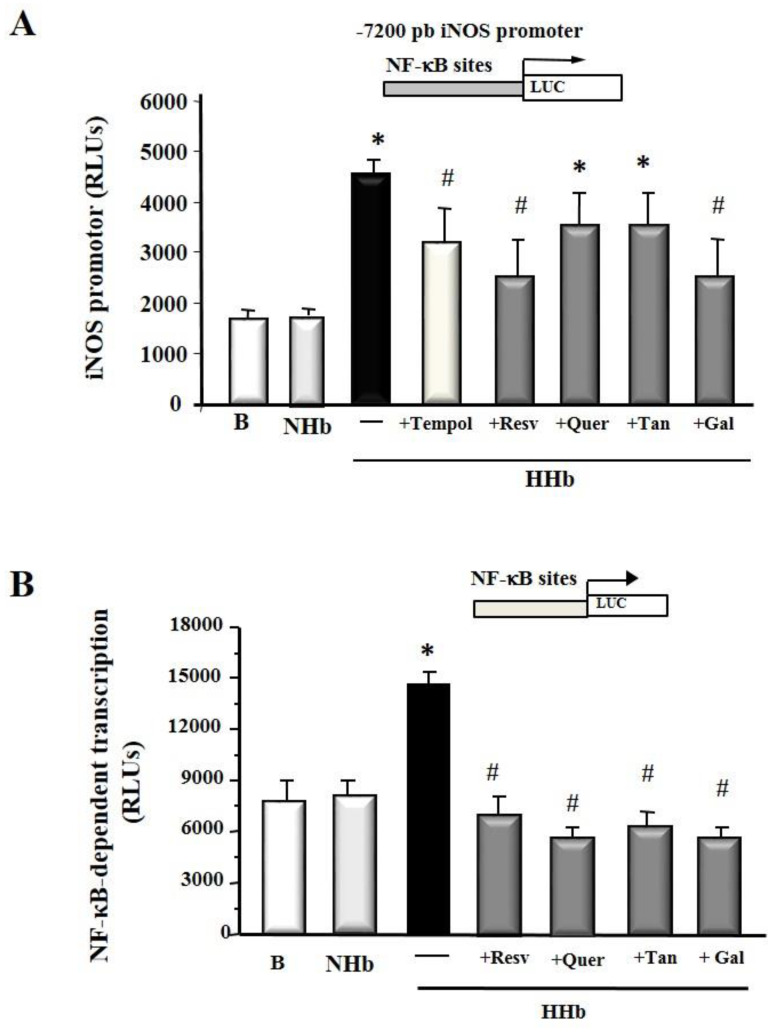
Polyphenols decrease proinflammatory gene expression induced by Amadori products. (**A**) The Human inducible Nitric Oxide synthetase (iNOS) promoter was studied using luciferase-based reporter plasmids transiently transfected in HPMCs; (**B**) nuclear factor-kappaB (NF-kB)-dependent transcriptional activation was assessed in cells transiently transfected with p5NF-kB-Luc plasmid. HPMCs were treated HHb and NHb (both at 10 nM); polyphenols: resveratrol (Resv, 12.5 μM), tannic acid (Tan, 10 μM), quercetin (Quer, 10 μM), gallic acid (Gal, 10 μM); or Tempol (Temp, 100 μM), a ROS scavenger. For each experiment, 10 independent cultures of HPMCs were used, corresponding to 10 donors. Data represent the mean ± SD of at least seven independent experiments (for each culture of HPMCs corresponding to each of the 10 donors) expressed as relative light units (RLUs). * *p* ≤ 0.05 vs. basal. # *p* ≤ 0.05 vs. HHb.

**Figure 3 antioxidants-09-00572-f003:**
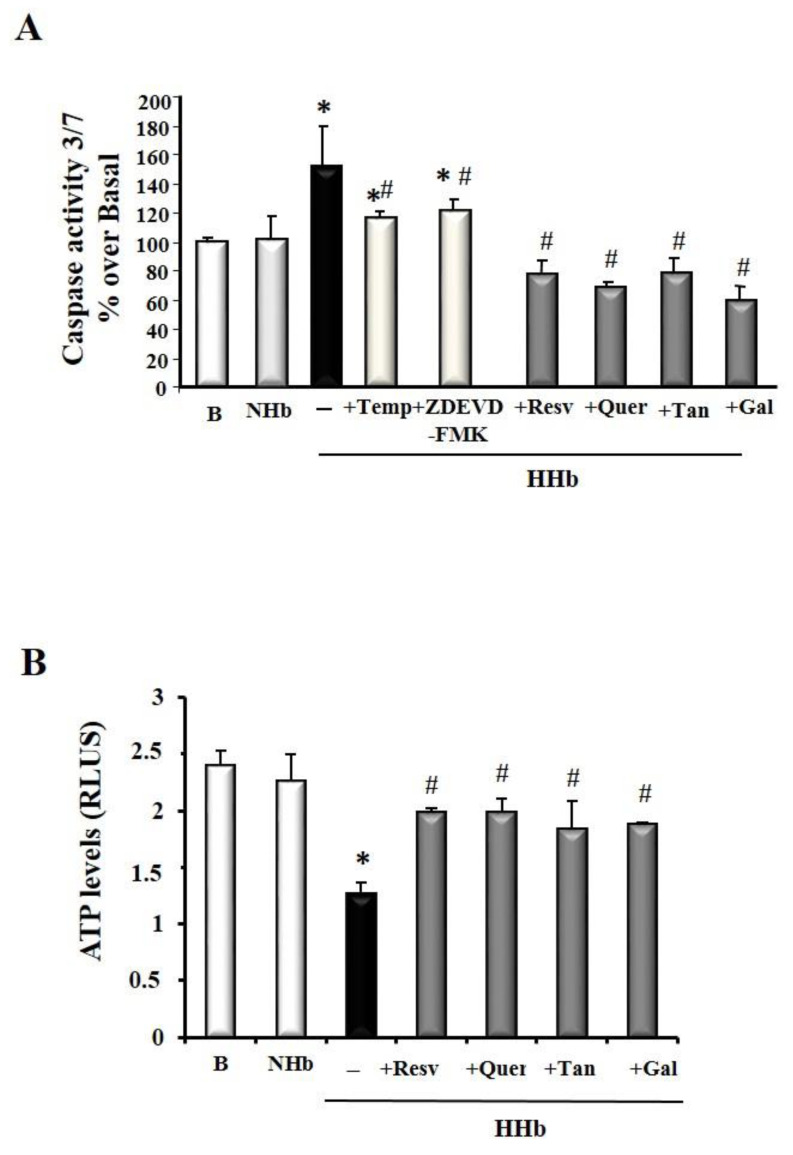
Polyphenols inhibit apoptosis induced by Amadori adducts in HPMCs. (**A**) Caspase 3/7 activity in whole cell lysates; (**B**) ATP levels measured by a luminescent assay in HPMCs. Cells exposed for 24 h to different Amadori adducts, HHb and NHb (10 nM), polyphenols: resveratrol (Resv, 12.5 μM), tannic acid (Tan, 10 μM), quercetin (Quer, 10 μM), gallic acid (Gal, 10 μM); Tempol (Temp, 100 μM) a ROS scavenger and ZDEVD-FMK (100 μM), a specific caspase-3 inhibitor. For each experiment, 10 independent cultures of HPMCs were used, corresponding to 10 donors. Data represent means ± SD of 7 independent experiments (for each culture of HPMCs corresponding to each of the 10 donors), (**A**) in % and (**B**) in relative light units, RLUs, * *p* ≤ 0.05 vs. el basal. # *p* ≤ 0.05 vs. HHb.

**Figure 4 antioxidants-09-00572-f004:**
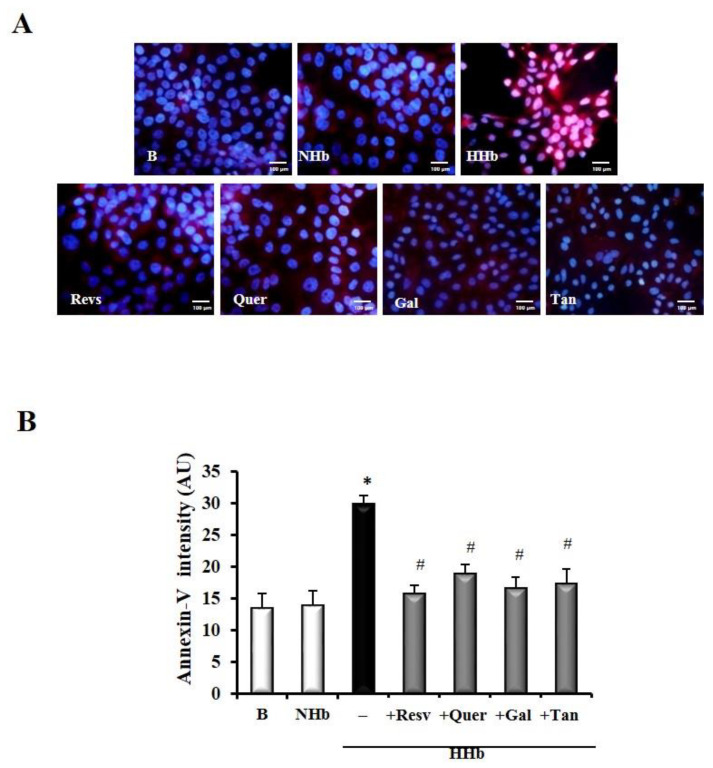
Levels of annexin V decreased associated with Polyphenols treatment in HPMC. (**A**) Representative immunostaining of annexin V in HPMCs. The cytoplasm shows annexin-positive staining (red) characteristic of apoptotic cells (×40). Scale bar 100 μm; (**B**) Quantification of fluorescence intensity for Annexin-V. Cells exposed for 24 h to different Amadori adducts, HHb and NHb (10 nM), polyphenols: resveratrol (Resv, 12.5 μM), tannic acid (Tan, 10 μM), quercetin (Quer, 10 μM), gallic acid (Gal, 10 μM). For each experiment, 10 independent cultures of HPMCs were used, corresponding to 10 donors. Data represent means ± SD of 7 independent experiments (for each culture of HPMCs corresponding to each of the 10 donors), (**B**) in Arbitrary units (AU), * *p* ≤ 0.05 vs. el basal. # *p* ≤ 0.05 vs. HHb.

**Figure 5 antioxidants-09-00572-f005:**
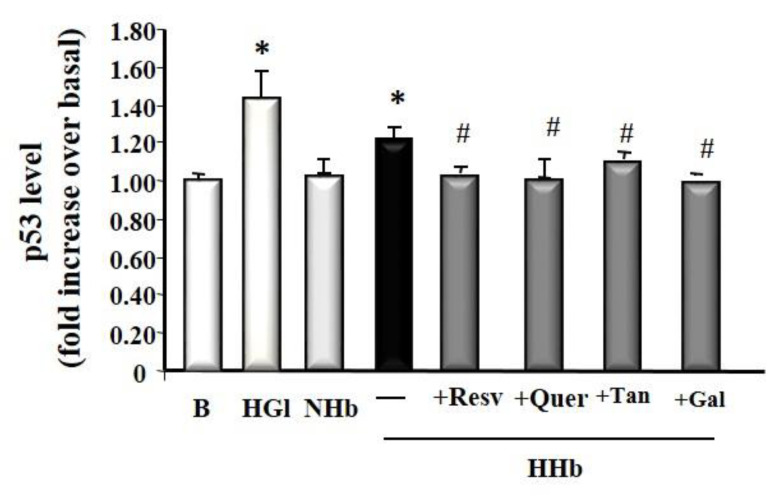
Polyphenols decrease tumour suppressor protein p53 levels induced by Amadori Adducts. Cells exposed for 24 h to different Amadori adducts, HHb and NHb (10 nM), polyphenols: resveratrol (Resv, 12.5 μM), tannic acid (Tan, 10 μM), quercetin (Quer, 10 μM), gallic acid (Gal, 10 μM); and 25 mM glucose (HGl, High Glucose). For each experiment, 10 independent cultures of HPMCs were used, corresponding to 10 donors. Data represent means ± SD of 7 independent experiments (for each culture of HPMCs corresponding to each of the 10 donors), * *p* ≤ 0.05 vs. el basal. # *p* ≤ 0.05 vs. HHb.

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
