# Peer review of "Polyphenols Attenuate Highly-Glycosylated Haemoglobin-Induced Damage in Human Peritoneal Mesothelial Cells"

_antioxidants, 2020, doi:10.3390/antiox9070572_

Round 1
Reviewer 1 Report
Overall a well written interesting study which is important for nutritional advice to patients. I had no issues with the paper except for the treatment of antioxidant effect being promoted as the mechanism for polyphenol where they have different effects, there is clearly more going on here and I have commented below. This is not a significant problem for the paper, and simply not citing the free radical theory as the global explanation would better reflect their actual data and be more in line with what a lot of the research into the actions of the polyphenols used in this study are showing.
Abstract
Clear and well written
Introduction
A well-developed introduction and review that covered the key issues. I am surprised that the authors consider that the main action for polyphenols in this case are as antioxidants. Polyphenols do a great deal more than supposedly reduce free radicals, they actively modify enzymatic and cellular functions, gene expression and often work more like drugs than free radical scavengers.
Results and discussion
The authors actually mention this in the results section ‘Polyphenols decrease reactive oxygen species and proinflammatory gene expression induced by Amadori products’. However, why are they stating the reduction in oxidative species inhibits gene expression, if this was the mechanism than any antioxidant would have the same effect. Instead different polyphenols have different effects on gene expression. It would be more logical to propose that polyphenols may affect gene expression and the antioxidant stabilisation effect on free radicals while important is secondary or parallel. I am not sure why the authors haven’t suggested this as they actually make a good case for it when discussing mechanisms in their results. The authors go on to say that polyphenols may exert anti-inflammatory effects through molecular mechanisms such as inhibition of enzymes… precisely the point I am alluding too… there is a lot more going on than antioxidant free radical scavenging. While the author cover the important issues, to me the continued reference back to antioxidants as being the global mechanism limits how the effects of these polyphenols should be understood. E.g. the authors discuss ‘Quercetin-induced cytoprotection, potentially based on its antioxidant effects in this model’…. Once again if everything works by an antioxidant effect then why do we see differences between polyphenols as they are all antioxidants? Quercetin has many actions including inducing autopsis in senescent cells, anti-inflammatory and perhaps improves autophagy.
While I understand being wedded to the antioxidant theory of everything I don’t think the science supports this is the case, and I actually think the authors have demonstrated this by their results. The solution is simple and all that is needed is not to reference everything back to some sort of global antioxidant affect as this confuses things. They have made a good case where antioxidant actions occur and where other things are going on. I think their analysis stands on its own in the paper.
Author Response
Dear Reviewer,
Thank you for your comments, below I answer your questions.
Comments and Suggestions for Authors:
Overall a well written interesting study which is important for nutritional advice to patients. I had no issues with the paper except for the treatment of antioxidant effect being promoted as the mechanism for polyphenol where they have different effects, there is clearly more going on here and I have commented below. This is not a significant problem for the paper, and simply not citing the free radical theory as the global explanation would better reflect their actual data and be more in line with what a lot of the research into the actions of the polyphenols used in this study are showing.
Response 1: Amended. Discussion has been changed to included different effects of the polyphenols in HPMCs.
Abstract:
Clear and well written
Introduction:
A well-developed introduction and review that covered the key issues. I am surprised that the authors consider that the main action for polyphenols in this case are as antioxidants. Polyphenols do a great deal more than supposedly reduce free radicals, they actively modify enzymatic and cellular functions, gene expression and often work more like drugs than free radical scavengers.
Results and discussion:
The authors actually mention this in the results section ‘Polyphenols decrease reactive oxygen species and proinflammatory gene expression induced by Amadori products’. However, why are they stating the reduction in oxidative species inhibits gene expression, if this was the mechanism than any antioxidant would have the same effect. Instead different polyphenols have different effects on gene expression. It would be more logical to propose that polyphenols may affect gene expression and the antioxidant stabilisation effect on free radicals while important is secondary or parallel. I am not sure why the authors haven’t suggested this as they actually make a good case for it when discussing mechanisms in their results. The authors go on to say that polyphenols may exert anti-inflammatory effects through molecular mechanisms such as inhibition of enzymes… precisely the point I am alluding too… there is a lot more going on than antioxidant free radical scavenging. While the author cover the important issues, to me the continued reference back to antioxidants as being the global mechanism limits how the effects of these polyphenols should be understood. E.g. the authors discuss ‘Quercetin-induced cytoprotection, potentially based on its antioxidant effects in this model’…. Once again if everything works by an antioxidant effect then why do we see differences between polyphenols as they are all antioxidants? Quercetin has many actions including inducing autopsis in senescent cells, anti-inflammatory and perhaps improves autophagy.
While I understand being wedded to the antioxidant theory of everything I don’t think the science supports this is the case, and I actually think the authors have demonstrated this by their results. The solution is simple and all that is needed is not to reference everything back to some sort of global antioxidant affect as this confuses things. They have made a good case where antioxidant actions occur and where other things are going on. I think their analysis stands on its own in the paper.
Response 2: Amended. Discussion has been changed to included different effects of the polyphenols in HPMCs.
Reviewer 2 Report
The manuscript “Polyphenols attenuates Amadori adducts-induced damage in human peritoneal mesothelial cells” by Carolina Sánchez-Rodríguez and Julián Nevado study the effect of treatment with polyphenols on oxidative stress, apoptosis and inflammation induced by Amadori adducts in Human Peritoneal Mesothelial Cells (HPMCs). The study is interesting and generally well done. In my opinion, however, some issue may be improved.
- Overall, the manuscript needs a thorough grammatical check, as there are numerous small mistakes in English usage and style (i.e. “galic acid” instead of “gallic acid” or “RVS” instead of “RSV” or “Quercitine” instead of “Quercetin”, in figure 2 legend, etc.).
- Polyphenols is a huge class of molecules with a broad spectrum of biological activities; therefore, the authors should provide more information about their choice to focus on tannic acid, gallic acid, quercetin and resveratrol, highlighting their healthy properties.
- The authors should present their results more clearly. For example, in the result section 3.3, the authors described only the effect of quercetin on the ATP levels, while in the 3.4 section they focused the attention on the resveratrol. The figure 2, instead, showed that all polyphenols exhibit similar effects.
Author Response
Dear Reviewer,
Thank you for your comments, below I answer your questions.
Point 1: Overall, the manuscript needs a thorough grammatical check, as there are numerous small mistakes in English usage and style (i.e. “galic acid” instead of “gallic acid” or “RVS” instead of “RSV” or “Quercitine” instead of “Quercetin”, in figure 2 legend, etc.).
Response 1: Amended, in the revised manuscript we made a grammatical check
Point 2: Polyphenols is a huge class of molecules with a broad spectrum of biological activities; therefore, the authors should provide more information about their choice to focus on tannic acid, gallic acid, quercetin and resveratrol, highlighting their healthy properties.
Response 2: In this study, we assessed the effects of common, dietary, natural polyphenols in HPMCs. The choice of polyphenols used in the study was based on the literature reviewed, focusing on those polyphenols with anti-inflammatory, anti-oxidative, and anti-glycation activities. Find bibliographic references and further explanation in the revised manuscript in the introduction and methods.
Point 3: The authors should present their results more clearly. For example, in the result section 3.3, the authors described only the effect of quercetin on the ATP levels, while in the 3.4 section they focused the attention on the resveratrol. The figure 2, instead, showed that all polyphenols exhibit similar effects.
Response 3: Amended, modified in the revised manuscript in the results.
Reviewer 3 Report
Dear Authors, Dear Editor,
The manuscript is very well written and conceived. It is important both from a scientific and applied point of view, being a first signal of the importance of natural polyphenols in the balanced diet of the patient and, moreover, of the patient with nephropathy. In vitro studies certify the use of vegetables, fruits or even pharmaceutical preparations with polyphenols. And future studies on human patients will bring valuable additional information.
Some observations to correct
Quercetin, Resveratrol, Gallic and Tannic Acid - in lower case
POLYPHENOLS ATTENUATES (in title) and Moringaoleifera - to be corrected
243 - Gallic acid - in lower case
250 - Quercetin - in lower case
Silybum marianum - italic
How did you choose the polyphenols you studied? Please explain this. Instead, could standardized plant extracts be used in polyphenols (caffeic acid, rosmarinic acid, rutin, etc.)?
Author Response
Dear Reviewer,
Thank you for your comments, below I answer your questions.
Point 1: Quercetin, Resveratrol, Gallic and Tannic Acid - in lower case
Response 1: Amended in the revised manuscript.
Point 2: POLYPHENOLS ATTENUATES (in title) and Moringaoleifera - to be corrected
Response 2: Amended in the revised manuscript.
Point 3: 243 - Gallic acid - in lower case
Response 3: Modified in the revised manuscript
Point 4: 250 - Quercetin - in lower case
Response 4: Modified in the revised manuscript.
Point 5: Silybum marianum - italic
Response 5:Modified in the revised manuscript.
Point 6: How did you choose the polyphenols you studied? Please explain this. Instead, could standardized plant extracts be used in polyphenols (caffeic acid, rosmarinic acid, rutin, etc.)?
Response 6: The choice of polyphenols used in the study was based on the literature reviewed, focusing on those polyphenols with anti-inflammatory, anti-oxidative, and anti-glycation activities. Find bibliographic references and further explanation in the revised manuscript in the introduction and methods.
Of course, extracts of plants rich in polyphenols can be used, our group previously studied the effect of extract of Ginkgo biloba leaves, EGb761, on age-associated cochlear caspase activation (Nevado et al., 2010). EGb761 treatment has a significant benefit with an early and preventive effect, reversing the deleterious effect of aging in the integrity of the rat cochlea, even in the late stage of the rat lifespan.
The major constituents of EGb 761 (0.1%) are: flavonol monoglycosides (eg, quercetin-3-0-glucoside, quercetin-3-0-rhamnoside, and 38-0-methylmyricetin-3-0-glucoside), flavonol diglycosides, flavonol triglycosides, coumaric esters of flavonol diglycosides, flavonoidic compound, terpenes (e.g, bilobalide, ginkgolides A, B, C, and J), organic acids, and steroids. (Diamond B., 2000).
Many other studies investigated the beneficial effect on health of different extracts rich in polyphenols, e.g, Li et al. 2018; Su et al., 2019; Anhê et al., 2017 or Varoni et al., 2012.
References:
Nevado J, Sanz J, Sánchez-Rodríguez C, García-Berrocal JR, Martín-Sanz E, González-García JA, Esteban-Sánchez J and Ramírez-Camacho R. Ginkgo biloba extract (EGb761) protects against aging-related caspase-mediated apoptosis in rat cochlea. Acta Oto-Laryngologica, 2010; 130: 1101–1112.
Diamond BJ, Shiflett SC, Feiwel N, Matheis RJ, Noskin O, Richards JA, Schoenberger NE. Ginkgo biloba extract: mechanisms and clinical indications. Arch Phys Med Rehabil. 2000;81(5):668-78.
Li XW, Chen HP, He YY, Chen WL, Chen JW, Gao L, Hu HY, Wang J. Effects of Rich-Polyphenols Extract of Dendrobium loddigesii on Anti-Diabetic, Anti-Inflammatory, Anti-Oxidant, and Gut Microbiota Modulation in db/db Mice. Molecules. 2018;23(12):3245.
Su X, Liu K, Xie Y, Zhang M, Wang Y, Zhao M, Guo Y, Zhang Y, Wang J. Protective effect of a polyphenols-rich extract from Inonotus Sanghuang on bleomycin-induced acute lung injury in mice. Life Sci. 2019;230:208-217.
Anhê FF, Nachbar RT, Varin TV, Vilela V, Dudonné S, Pilon G, Fournier M, Lecours MA, Desjardins Y, Roy D, Levy E, Marette A. A polyphenol-rich cranberry extract reverses insulin resistance and hepatic steatosis independently of body weight loss. Mol Metab. 2017;6(12):1563-1573.
Varoni EM, Lodi G, Sardella A, Carrassi A, Iriti M. Plant polyphenols and oral health: old phytochemicals for new fields. Curr Med Chem. 2012;19(11):1706-20.
Reviewer 4 Report
This paper by Sánchez-Rodríguez and Nevado showed demonstrated that the effect of dietary polyphenols on Amadori-adducts-induced cell death in Human Peritoneal Mesothelial Cells (HPMCs). They also showed that these dietary polyphenols decrease intracellular reactive oxygen species level and proinflammatory gene expression induced by Amadori products. From the results obtained in these investigations, the authors concluded that dietary polyphenols might have a translational potential in patients receiving peritoneal dialysis. However, I feel that the authors do not have sufficient data to make this conclusion. Especially, I fell that the Discussion about the cytoprotective mechanize of dietary polyphenols is not convincing.
Major point
1. Statistical analysis L129-131
You described that Statistical significance (p<0.05) was evaluated by unpaired Student’s “t” test or factorial analysis of variance (ANOVA), as required, using the StatView statistics program. Where did you use Student’s “t” test? Moreover, you should state post hoc test in ANOVA.
2. NOS Data
Although the author showed that iNOS promoter activity was recovery after the treatment of polyphenols, I strongly recommend you to conduct the level of iNOS protein and active dimer by western blots. I wonder the other isoforms of NOS, eNOS and iNOS, were no change or non-expression in the HPMCs. The authors should explain this point in the revised manuscript.
3.Concentration
The dose-response relationship is an important property in pharmacological research. Therefore, the authors show the data at some dose of polyphenols in vitro. Otherwise, the authors should explain why you used this concentrations in the revised manuscript.
4. Discussion (L233-245)
This paragraph is not convincing. In the discussion section, the authors are supposed to explain the data with comparison to previous findings in the discussion. The feasible underlying mechanism should be also discussed on the basis of experimental evidence but not of speculations here. In principle, the discussion must be based on the evidence presented. Accordingly, the discussion should be adequately revised.
Minor point
1 Resutls l196,
Fig 4 is lacking. Please correct Figure number in this sentence.
Author Response
Dear Reviewer,
Thank you for your comments, below I answer your questions.
Point 1. Statistical analysis L129-131
You described that Statistical significance (p<0.05) was evaluated by unpaired Student’s “t” test or factorial analysis of variance (ANOVA), as required, using the StatView statistics program. Where did you use Student’s “t” test? Moreover, you should state post hoc test in ANOVA.
Response 1: Student's “t” test was not used for statistical analysis; it was a transcription error when writing the manuscript. It has been corrected in revised manuscript. Thank you for your comment.
Point 2. NOS Data
Although the author showed that iNOS promoter activity was recovery after the treatment of polyphenols, I strongly recommend you to conduct the level of iNOS protein and active dimer by western blots. I wonder the other isoforms of NOS, eNOS and iNOS, were no change or non-expression in the HPMCs. The authors should explain this point in the revised manuscript.
Response 2: Indeed, to complete the work it would have been good to determine by Western Blot the levels of protein iNOs and their isoforms, but we focused on transfection assays with genes to determine their activity. NOS and iNOS isoforms, they were determined by our group in the previous study Nevado et al., 2005. The results we obtained show that NOS and iNOS activity increases in the presence of Hhb (10nM) but eNOS is not modified (data not show). Therefore, in this study we did not determine the activity of eNOS, although we could have measured the effect of polyphenols on the activity of NOS, but we focused more on determining its effect on the expression of NF-kB related proinflammatory genes, such as iNOS.
Nevado J, Peiró C, Vallejo S, El-Assar M, Lafuente N, Matesanz N, Azcutia V, Cercas E, Sánchez-Ferrer CF, Rodríguez-Mañas L. Amadori adducts activate nuclear factor-kappaB-related proinflammatory genes in cultured human peritoneal mesothelial cells. Br J Pharmacol. 2005;146(2):268-79.
Point 3. Concentration
The dose-response relationship is an important property in pharmacological research. Therefore, the authors show the data at some dose of polyphenols in vitro. Otherwise, the authors should explain why you used this concentrations in the revised manuscript.
Response 3: I agree with you that the dose-response relationship is very important in pharmacological research, therefore, for the choice of doses of the polyphenols used in the study, the literature was reviewed, focusing on in vitro studies. The polyphenols proves inhibitory to either a huge panoply of molecular targets in the micromolar concentration range, by down-regulating or suppressing many inflammatory pathways and functions.
Find bibliographic references and further explanation in the revised manuscript in the methods.
Reference:
Mrvová N, Skandík M, Kuniaková M and Racková L. Modulation of BV-2 microglia functions by novel quercetin pivaloyl ester. Neurochemistry International. 2015;90: 246–254.
Ramyaa P, Krishnaswamy R, and Padma VV. Quercetin modulates OTA-induced oxidative stress and redox signaling in HepG2 cells—up regulation of Nrf2 expression and down regulation of NF-?B andCOX-2. Biochimica et Biophysica Acta (BBA)—General Subjects. 2014; 1840(1):681–692.
Ferruelo A, Romero I, Cabrera PM, Arance I, Andrés G, Angulo JC. Effects of resveratrol and other wine polyphenols on the proliferation, apoptosis and androgen receptor expression in LNCaP cells. Actas Urol Esp. 2014;38(6):397-404.
Ferruelo A, de Las Heras MM, Redondo C, Ramón de Fata F, Romero I, Angulo JC. Wine polyphenols exert antineoplasic effect on androgen resistant PC-3 cell line through the inhibition of the transcriptional activity of COX-2 promoter mediated by NF-kβ. Actas Urol Esp. 2014;38(7):429-37.
Ma Ch, Wang Y, Dong L, Li M, Cai W. Anti-inflammatory effect of resveratrol through the suppression of NF-κB and JAK/STAT signaling pathways. Acta Biochimica et Biophysica Sinica, Volume 47, Issue 3, March 2015, Pages 207–213.
Li Y, Yao J, Han C, Yang J, Chaudhry MT, Wang S, Liu H, Yin Y. Quercetin, Inflammation and Immunity. Nutrients. 2016 Mar 15;8(3):167.
Point 4. Discussion (L233-245)
This paragraph is not convincing. In the discussion section, the authors are supposed to explain the data with comparison to previous findings in the discussion. The feasible underlying mechanism should be also discussed on the basis of experimental evidence but not of speculations here. In principle, the discussion must be based on the evidence presented. Accordingly, the discussion should be adequately revised.
Response 4: The discussion has been reviewed and modified following the indications of your comments.
Point 5. Minor point, 1 Resutls l196, Fig 4 is lacking. Please correct Figure number in this sentence.
Response 5: Modified in the revised manuscript
Reviewer 5 Report
The manuscript reports an interesting study aimed to determine the cytoprotective role of the dietary polyphenols on damage induced by Amadori adducts in Human Peritoneal Mesothelial Cells. This paper is well-written and presents an interesting and carefully designed research. Some minor issues should be resolved before publishing this paper.
Page 2, line 57 - Authors should provide more information on the mechanism of action of polyphenols and their interactions with other compounds.
Page 2, lines 62-65 - This sentence should be moved to abstract.
Page 3, lines 96 - Please provide references that were used in the development of the method for the quantitation of ATP-levels.
Page 3, lines 102 - Please provide references that were used in the development of the method for the assaying annexin V binding.
The author contributions section is inconsistent with the requirements of Antioxidants, please check and correct.
Author Response
Dear Reviwer,
Thank you for your comments, below I answer your questions.
Point 1. Page 2, line 57 - Authors should provide more information on the mechanism of action of polyphenols and their interactions with other compounds.
Response 1: Amended, Information on the mechanisms of action of polyphenols are explained in the revised manuscript, in the Introduction.
In the case of interactions with other compounds, bioavailability and activity of polyphenols depend on foods’ structure and interactions with other food constituents, especially proteins, lipids, and carbohydrates. Polyphenols–proteins interactions can result in various biological effects, such as sense of astringency (Ferrer-Gallego et al., 2015). So far, polyphenols interactions with food lipids have not been of special importance, except in case of plant oils. Polyphenols–carbohydrates interactions can influence the organoleptic properties, while interactions with dietary fibers are particularly significant. Polyphenols can decrease the synthesis of fats and fatty acids in the liver, or delay their absorption in intestines. Also, polyphenols can slow down digestion of carbohydrates, through the inhibition of digestive enzymes or modulation of glucose uptake (Kardum and Glibetic, 2018).
Both animal and plant proteins have low impact on the bioavailability of polyphenols, but some in vitro studies reported that milk proteins could enhance intestinal absorption of polyphenols from tea. Dietary fats may alter the passage of polyphenols through gastrointestinal tract and impact absorption of more hydrophobic polyphenols in particular. While some studies reported that associations with carbohydrates could decrease bioavailability of polyphenols, the others showed the opposite effects. Macronutrients can be used for encapsulation of polyphenols, which can increase their bioavailability and ensure controlled and targeted release. Polyphenols’ interactions in the body include their incorporation in cell membranes which causes changes in fatty acid profile and impacts membrane-bound transporters and enzymes. Finally, gut microbiota plays essential role in metabolism of both polyphenols and macronutrients and thus can have great impact on their interactions (Holmes et al., 2012; Possemiers, Bolca, Verstraete, and Heyerick, 2011; Duda-Chodak, Tarko, Satora, and Sroka, 2015; Queipo-Ortuño et al., 2012).
Point 2. Page 2, lines 62-65 - This sentence should be moved to abstract.
Response 2: Modified in the revised manuscript
Point 3. Page 3, lines 96 - Please provide references that were used in the development of the method for the quantitation of ATP-levels.
Response 3: The references used in the development of the method for the quantification of ATP levels were based on previous studies from our and other groups, the following references have been included in the revised manuscript:
Nevado J, Sanz R, Casqueiro JC, Ayala A, García-Berrocal JR, Ramírez-Camacho R. Ageing evokes an intrinsic pro-apoptotic signalling pathway in rat cochlea. Acta Otolaryngol.2006; 126: 1134–1139.
García-Berrocal JR, Nevado J, Ramírez-Camacho R, Sanz R, González-García JA, Sánchez-Rodríguez C, Cantos B, España P, Verdaguer JM 5 and A Trinidad Cabezas. The anticancer drug cisplatin induces an intrinsic apoptotic pathway inside the inner ear. British Journal of Pharmacology. 2007; 152:1012–1020.
Point 4. Page 3, lines 102 - Please provide references that were used in the development of the method for the assaying annexin V binding.
Response 4: The references used in the development of the method for the quantification of annexin V levels were based on previous studies from our and other groups, the following references have been included in the revised manuscript:
Benevolensky D, Belikova Y, Mohammadzadeh R, Trouvé P, Marotte F, Russo-Marie F, Samuel JL, Charlemagne D. Expression and localization of the annexins II, V, and VI in myocardium from patients with end-stage heart failure. Lab Invest. 2000;80(2):123-33.
Sánchez-Rodríguez C, Peiró C, Vallejo S, et al. Pathways Responsible for Apoptosis Resulting from Amadori-Induced Oxidative and Nitrosative Stress in Human Mesothelial Cells. Am J Nephrol. 2011; 34:104–114
Point 5. The author contributions section is inconsistent with the requirements of Antioxidants, please check and correct.
Response 5: The author contributions section has been corrected according to the Antioxidants requirements in the revised manuscript.
Round 2
Reviewer 2 Report
The authors have made a good effort in addressing the issues raised in the previous round.
Reviewer 4 Report
The manuscript revised by Sánchez-Rodríguez et al has been well-corrected according to the comments from the reviewers. Some points are still remained to revise in the manuscript.
In Statistical analysis (178-181)
What kind of post hoc tests are you using in this study? Please described that in the revised manuscript.
Fig.1
Author should be check the SD in figure 1 again. Did you show that in Fig 1?
Fig. 4A
The quality of Immunofluorescent staining within Gal-treated cells is poor. The author should replace the original image to a better image in the revised manuscript. I feel the representative image did not match the graph in fig.4 B.